# Temporal trends of hospitalizations, comorbidity burden and in-hospital outcomes in patients admitted with asthma in the United States: Population-based study

Salwa S. Zghebi[1,2]*, Mohamed O. Mohamed[3,4], Mamas A. Mamas[3], Evangelos Kontopantelis[1,5]

1 Division of Population Health, Health Services Research and Primary Care, School of Health Sciences, Faculty of Biology, Medicine and Health, Manchester Academic Health Science Centre (MAHSC), The University of Manchester, Manchester, United Kingdom, 2 Department of Pharmaceutics, Faculty of Pharmacy, University of Tripoli, Tripoli, Libya, 3 Keele Cardiovascular Research Group, Centre for Prognosis Research, School of Primary, Community and Social Care, Keele University, Keele, United Kingdom, 4 Institute of Health Informatics, University College London, London, United Kingdom, 5 Division of Informatics, Imaging and Data Sciences, School of Health Sciences, Faculty of Biology, Medicine and Health, Manchester Academic Health Science Centre (MAHSC), The University of Manchester, Manchester, United Kingdom

* salwa.zghebi@manchester.ac.uk

## Abstract

### Background

Asthma is a prevalent chronic respiratory condition and remains a common cause for hospitalization. However, contemporary data on asthma hospitalization rates, comorbidity burden, and in-hospital outcomes are lacking.

### Methods

Survey-weighted analysis of hospitalization records with a primary diagnosis of asthma using data from the US National (Nationwide) Inpatient Sample between 2004 and 2017. Outcomes were number of hospitalizations per 100,000 population and in-hospital outcomes including receipt of ventilation, length of stay, and hospital costs. Patient and admission characteristics and comorbidity burden were examined over time. Multivariable logistic and linear regression models were fitted for over-time risks of the outcomes.

### Results

Among 3,098,863 asthma admissions between 2004 and 2017, mean (±SD) age was 29 (±25), 57% females, 36% White, 40% had Medicaid as primary payer. During 2004–2017, asthma hospitalizations declined from 89 to 56 per 100,000 population; length of stay remained overall stable; median (interquartile range IQR) inflation-adjusted hospital costs doubled from $8,446 (9,227) in 2004 to $17,756 (19,434) in 2017. Common comorbidities in patients admitted with asthma were hypertension and diabetes in adults, but gastroesophageal reflux disease, obstructive sleep apnoea, anemia, and obesity in children. Over time,

Project (HCUP) - National (Nationwide) Inpatient Sample (NIS) via their website https://www.hcup-us.ahrq.gov/nisoverview.jsp. All interested researchers can access the data through HCUP directly and we are not permitted to share the data or make it available as per the data use agreement with HCUP. The authors did not have any special access privileges to this data.

**Funding:** This study is funded by The University of Manchester, Manchester, UK, as part of the Presidential Fellowship provided to SSZ. The funders had no role in study design, data collection and analysis, decision to publish, or preparation of the manuscript.

**Competing interests:** The authors have declared that no competing interests exist.

the prevalence of mental illness increased by >50%. Severe asthma (IRR, 2.48; 95%CI: 2.27–2.72) and psychoses (IRR, 1.10; 1.05–1.14) were predictors of prolonged hospitalization. Asian/Pacific Islanders were more likely to receive ventilation (OR: 2.35; 1.73–3.20) than White patients. Hospital costs were significantly higher in females and adults with hypertension (coefficient, 1405.2; 283.1–2527.4) or psychoses (coefficient, 1978.4; 674.9–3282.0).

## Conclusions

US asthma hospitalization rates fluctuated in earlier years but declined over time, which may reflect improvements in community care and declining asthma prevalence. Comorbidity burden, including mental illness, increased over time and is associated with in-hospital outcomes. This highlights the changing landscape of asthma admissions which may inform redesigning services to support pre-hospitalization asthma care and help further reduce admissions, particularly among patients with multimorbidity.

## Introduction

Asthma is a common chronic respiratory condition affecting more than 25 million people in the USA, corresponding to nearly 7.7% and 8.4% of adults and children, respectively, and accounted for 479,300 hospitalizations in 2009 [1–4]. The Centers for Disease Control and Prevention (CDC) estimates the percentage of US population with asthma increased from 7.3% to 7.9% between 2001 and 2017 [4].

With ageing population, people with a long-term condition often develop multimorbidity (the co-occurrence of ≥2 chronic conditions, known as comorbidities) [5, 6]. Certain comorbidities are prevalent among people with asthma, including mental illness, chronic sinusitis, obstructive sleep apnoea (OSA), atopic dermatitis, and gastroesophageal reflux disease (GERD) which reportedly affect asthma control, outcomes, and healthcare utilization [7, 8]. The impact of multimorbidity is known to be greater than the additive effect of multiple individual conditions and is associated with poorer quality of life, adverse outcomes, and greater burden on national healthcare recourses [5, 9]. Some past studies examined the trends of asthma hospitalizations and outcomes up to 2011 using the US National (Nationwide) Inpatient Sample (NIS) [1, 3, 10], there are however limited contemporary data on the patient characteristics and comorbidity burden in patients admitted with asthma and/or their changes over time. Previous studies either did not account for comorbidities in the analyses [11] and/or reported most findings for the whole cohort and not on an annual basis [1, 3], not providing trends of data over time. Mapping comorbidity burden is key to understand which conditions coexist in hospitalized people with asthma, and to examine if comorbidity phenotype changes over time by patient factors, and whether different comorbidities have different prognostic impacts.

Using data from the US National (Nationwide) Inpatient Sample (NIS) of patients admitted with asthma between 2004 and 2017, we aimed to: 1) examine annual trends of hospitalizations and sociodemographic factors; 2) describe annual trends of comorbidity burden by patient age, sex, race and examine the association between predictors and comorbidities and in-hospital outcomes over time.

## Methods

**Data source.** Retrospective cohort study using data from the US National (Nationwide) Inpatient Sample (NIS) between January 2004 and December 2017. The NIS is sponsored by the Agency for Healthcare Research and Quality (AHRQ) and is the largest available all-payer data on inpatient stays from all US states participating in the Healthcare Cost and Utilization Project (HCUP), covering >97% of the US population [12]. NIS design approximates a 20% stratified sample of all admissions from community hospitals, excluding rehabilitation, long-term acute care hospitals [12]. NIS provides anonymized information on primary and secondary hospitalization diagnoses from >7 million annual inpatient stays, recorded in typical discharge abstracts [13, 14]. AHRQ sampling weights for each admission were used to calculate national estimates. Modified weights were used in all analyses to account for the NIS sampling design change in 2012.

## Study population

Patients with a primary diagnosis of asthma were identified using the International Classification of Diseases (ICD)-Ninth Revision-Clinical Modification (ICD-9-CM) and ICD-Tenth Revision-Clinical Modification/Procedure Coding System (ICD-10-CM/PCS) codes (493*, J45*, J46). Patients were categorized by age into paediatric (0-17years) and adult (≥18years) groups. ICD-9-CM and ICD-10-CM/PCS codes were used before and from 01/10/2015, respectively [12].

## Outcome measures and variables

The primary outcome was weighted number of hospital admissions per 100,000 population. Secondary in-hospital outcomes were intubation/mechanical ventilation use, length of stay (LOS), and total hospital costs. Outcomes were examined longitudinally, and we also described patient characteristics and comorbidity burden annually over time.

Variables: admission year, age, sex, race (White, Black, Hispanic, Asian or Pacific Islander, Native American, Other, unknown), median household income (quartiles for patient's ZIP Code), patient's residence location, asthma severity (indicated using a categorical variable which describes the 'severity of illness subclass': No class specified, minor loss of function, moderate loss of function, major loss of function, extreme loss of function), primary payer (Medicare, Medicaid, private, self-pay, no charge, other, unknown), and 20 comorbidities (including HCUP Elixhauser comorbidity software) [15, 16]: cardiovascular disease (CVD) (peripheral vascular disease, heart failure, myocardial infarction, coronary heart disease, atrial fibrillation and flutter, heart valve disease, transient ischemic attack, and stroke); chronic obstructive pulmonary disease (COPD); dyslipidaemia; hypertension (HT); coagulopathy; anemia; diabetes; hypothyroidism; chronic kidney disease; liver disease; weight loss; psychoses; rheumatoid arthritis and vascular collagen disease; lung cancer or cancer; and comorbidities reportedly affect asthma control and outcomes [8]: chronic sinusitis, obstructive sleep apnoea (OSA), atopic dermatitis, gastroesophageal reflux disease (GERD), obesity, and depression. Admission factors included weekend admission, calendar month, admission quarter, elective admission, LOS, and total charges. Hospital-level factors included hospital bed size, hospital control/ownership, hospital location/teaching status, and region. Admission records with missing age, sex, admission year, LOS, or death status were excluded (N = 50,617, 7%). Missing race, income, residence location, primary payer, and hospital factors were assigned to a separate 'unknown' category.

## Data analysis

Categorical variables are described as percentages, and differences between group proportions were tested using Chi-squared test (over whole study period). Continuous variables are described as mean (standard deviation SD) or median (interquartile range IQR).

Baseline characteristics, admission rates (as number and rates per 100,000 population), trends of in-hospital deaths (overall and by patient factors), LOS, and total charges are reported over time. Annual charges are adjusted for inflation to 2017 costs based on the latest US government consumer price index data (12/01/2022) [17].

Multivariable regression models adjusted for age, sex, race, asthma severity, elective admission, weekend admission, admission quarter were used to assess the associations between comorbidities and in-hospital outcomes at study start (2004), midpoint (2010), and end (2017) by age groups (<18, ≥18 years). Age, sex, and race are key demographic predictors for in-hospital outcomes in patients with asthma as reported previously [1, 3, 10]. Asthma severity and elective admissions were included to adjust for differences in disease severity between patients [10]. Weekend admissions and admission quarter (mainly winter months) have been reported as predictors of in-hospital outcomes in US asthma admissions [1]. For intubation/ventilation use, logistic regression models were used to estimate odds ratio (OR) and 95% confidence interval (CI); linear regression was used for total charges; Poisson regression was used to estimate incidence rate ratio (IRR) and 95%CI for the LOS outcome. Poisson regression was chosen over negative binomial regression as the distribution of LOS was not widely dispersed when assessed on all study years. The included comorbidities were selected based on prevalence and strength of association with each outcome from yearly age-specific univariable models. All analyses were survey-weighted to produce a nationally-representative estimate of the entire US population of hospitalized patients. The database provider recommends a few approaches to assess the possible effect of the switch from ICD-9 to ICD-10 coding from 1st October 2015 on trends of diagnoses. We assessed the possible impact of the switch on the trends of asthma admissions by examining whether the patients' characteristics and the estimated admission rates differed before and after the code switch by calendar quarters. Analyses were conducted using Stata 16 [18]. The study is reported in accordance to the REporting of studies Conducted using Observational Routinely collected health Data (RECORD) guidelines checklist (an update of the STROBE guidelines) [19].

## Results

### Annual trends of asthma admissions

Overall, 3,098,863 weighted asthma admissions were recorded during 2004–2017 (Table 1, S1 Table). Annual admissions fluctuated between 64.4 and 91.6 per 100,000 population up to 2012 before declining from 2013 reaching 55.6/100,000 population in 2017. Admission rates ranged between 102 and 186/100,000 population in children (aged <18y); between 42 and 60 in adults; 46 and 82 in males; and between 65 and 101/100,000 population in females with asthma (Fig 1). Mean (±SD) age was 28.5(±25) years, 57% females, 36% White, 52% with minor vs. 1.4% with extreme loss of function (asthma severity levels), 4.3% of patients required intubation/ventilation, and 40% had Medicaid as primary payer.

Across the study period, the highest admissions were recorded between October-March whereas lowest admissions between July-September annually; 25% were on weekends, and mainly in large (56%) or urban/teaching hospitals (58%). LOS remained relatively unchanged (2.71±2.5 days), while median (IQR) inflation-adjusted total charges doubled from $8,446 (9,227) in 2004 to $17,756 (19,434) in 2017 and both notably increased with increasing asthma severity level (S1 Fig).

**Table 1. Annual trends of admissions and baseline characteristics of patients admitted with asthma between 2004 and 2017.**

| | Overall (2004–17) | 2004 | 2005 | 2006 | 2007 | 2008 | 2009 | 2010 | 2011 | 2012 | 2013 | 2014 | 2015 | 2016 | 2017 |
|---|---|---|---|---|---|---|---|---|---|---|---|---|---|---|---|
| Weighted admissions, N | 3,098,863 | 260,003 | 270,750 | 248,919 | 224,714 | 214,789 | 245,772 | 223,147 | 200,524 | 233,520 | 212,685 | 207,800 | 189,735 | 185,565 | 180,940 |
| Admissions per 100,000 population | 995.7 | 88.8 | 91.6 | 83.4 | 74.6 | 70.6 | 80.1 | 72.1 | 64.4 | 74.4 | 67.3 | 65.2 | 59.0 | 57.4 | 55.6 |
| Age, mean (±SD) | 28.5 (25) | 26.6 (25) | 26.6 (25) | 27.4 (25) | 28.0 (25) | 29.4 (25) | 27.8 (25) | 28.1(25) | 30.1(25) | 27.8 (25) | 29.0(26) | 27.9 (25) | 29.8 (26) | 31.3 (26) | 32.2 (26) |
| Female, % | 57 | 56 | 56 | 57 | 57 | 58 | 56 | 57 | 58 | 56 | 57 | 56 | 58 | 59 | 59 |
| In-hospital deaths, % | 0.11 | 0.10 | 0.08 | 0.09 | 0.11 | 0.08 | 0.09 | 0.11 | 0.11 | 0.10 | 0.12 | 0.17 | 0.14 | 0.17 | 0.18 |
| Ethnicity, % | | | | | | | | | | | | | | | |
| • White | 35.8 | 32.2 | 32.3 | 31.2 | 30.9 | 37.6 | 36.0 | 38.3 | 39.3 | 39.3 | 38.2 | 37.8 | 37.3 | 37.1 | 36.9 |
| • Black | 27.3 | 23.0 | 19.3 | 22.2 | 21.3 | 25.4 | 23.9 | 30.8 | 29.4 | 31.4 | 31.6 | 32.4 | 32.5 | 32.9 | 32.6 |
| • Hispanic | 15.6 | 14.7 | 11.2 | 16.1 | 13.7 | 12.1 | 17.5 | 16.2 | 13.8 | 17.2 | 17.8 | 17.7 | 18.0 | 17.8 | 18.7 |
| • Asian/ Pacific Islander | 2.0 | 1.7 | 1.5 | 1.3 | 1.4 | 1.5 | 2.0 | 2.4 | 1.70 | 2.2 | 2.4 | 2.3 | 2.6 | 2.5 | 3.0 |
| • Native American | 0.7 | 0.44 | 0.35 | 0.7 | 0.80 | 0.56 | 0.82 | 0.74 | 0.9 | 0.72 | 0.66 | 0.71 | 0.8 | 0.7 | 0.7 |
| • Other | 4.0 | 2.7 | 4.1 | 2.8 | 2.9 | 3.6 | 6.3 | 3.4 | 4.0 | 4.2 | 4.2 | 4.3 | 4.2 | 4.4 | 4.6 |
| • Unknown | 14.6 | 26.2 | 31.2 | 25.8 | 29.0 | 19.1 | 13.5 | 8.2 | 11.0 | 5.0 | 5.1 | 5.0 | 4.8 | 4.6 | 3.6 |
| Median household income (quartiles), % | | | | | | | | | | | | | | | |
| • $1-$43,999 | 31.0 | NA | NA | 38.2 | 36.5 | 38.55 | 36.5 | 35.1 | 36.0 | 37.4 | 36.4 | 36.8 | 39.4 | 39.4 | 39.4 |
| • $44,000-$55,999 | 19.3 | NA | NA | 23.2 | 23.6 | 24.71 | 22.5 | 23.0 | 22.3 | 23.1 | 23.7 | 24.5 | 21.7 | 23.2 | 23.2 |
| • $56,000–73,999 | 16.6 | NA | NA | 19.2 | 19.6 | 18.15 | 18.2 | 21.2 | 21.8 | 20.1 | 20.9 | 19.7 | 20.6 | 20.5 | 20.5 |
| • ≥ $74,000 | 13.7 | NA | NA | 16.6 | 16.3 | 17.13 | 16.5 | 17.6 | 17.7 | 16.5 | 16.2 | 15.7 | 16.1 | 15.7 | 15.7 |
| • Unknown | 19.5 | NA | NA | 2.9 | 4.1 | 1.47 | 6.3 | 3.1 | 2.3 | 2.9 | 2.9 | 3.2 | 2.2 | 1.2 | 1.2 |
| Asthma severity, % | | | | | | | | | | | | | | | |
| • Minor | 51.5 | 56.3 | 56.6 | 57.3 | 55.0 | 53.8 | 52.6 | 51.4 | 48.3 | 54.0 | 50.5 | 48.5 | 45.3 | 42.8 | 41.6 |
| • Moderate | 37.6 | 36.5 | 36.0 | 36.0 | 37.7 | 38.0 | 38.5 | 38.2 | 39.7 | 36.8 | 38.2 | 38.7 | 39.2 | 37.8 | 36.1 |
| • Major | 9.5 | 6.4 | 6.5 | 5.8 | 6.5 | 7.3 | 7.9 | 9.0 | 10.5 | 8.0 | 9.9 | 11.1 | 13.7 | 17.1 | 19.7 |
| • Extreme | 1.4 | 0.94 | 0.92 | 0.9 | 0.93 | 1.0 | 1.0 | 1.5 | 1.4 | 1.2 | 1.5 | 1.7 | 1.8 | 2.2 | 2.7 |
| Primary payer, % | | | | | | | | | | | | | | | |
| • Medicare | 15.0 | 12.5 | 13.4 | 13.4 | 13.7 | 15.0 | 13.4 | 13.8 | 16.3 | 14.8 | 16.3 | 15.0 | 16.9 | 18.5 | 19.7 |

(*Continued*)

**Table 1.** (Continued)

| | Overall (2004–17) | 2004 | 2005 | 2006 | 2007 | 2008 | 2009 | 2010 | 2011 | 2012 | 2013 | 2014 | 2015 | 2016 | 2017 |
|---|---|---|---|---|---|---|---|---|---|---|---|---|---|---|---|
| • Medicaid | 39.6 | 37.1 | 36.2 | 36.7 | 34.9 | 35.7 | 38.8 | 38.8 | 37.9 | 41.4 | 41.3 | 45.5 | 45.4 | 44.6 | 44.0 |
| • Private | 33.5 | 38.3 | 38.3 | 38.0 | 37.2 | 37.5 | 33.2 | 34.1 | 33.3 | 30.8 | 29.6 | 29.5 | 28.5 | 27.9 | 27.4 |
| • Self-pay | 8.1 | 8.6 | 8.1 | 8.1 | 9.8 | 8.1 | 10.7 | 9.2 | 8.3 | 8.5 | 8.4 | 6.7 | 6.0 | 6.0 | 6.1 |
| • No charge | 0.56 | 0.29 | 0.48 | 0.45 | 0.53 | 0.5 | 0.58 | 0.49 | 0.50 | 0.70 | 1.1 | 0.70 | 0.55 | 0.50 | 0.50 |
| • Other | 3.1 | 2.9 | 3.5 | 3.1 | 3.7 | 3.0 | 3.2 | 3.40 | 3.40 | 3.6 | 3.2 | 2.5 | 2.4 | 2.3 | 2.2 |
| • Unknown | 0.20 | 0.19 | 0.07 | 0.17 | 0.20 | 0.23 | 0.27 | 0.28 | 0.30 | 0.20 | 0.12 | 0.20 | 0.16 | 0.20 | 0.20 |
| Intubation/ ventilator, % | 4.27 | 3.66 | 3.04 | 3.52 | 2.9 | 3.25 | 3.1 | 3.74 | 3.7 | 4.4 | 5.02 | 6.10 | 6.32 | 6.16 | 6.71 |
| LOS (days), mean (±SD) | 2.71 (±2.5) | 2.79 (±2.7) | 2.75 (±2.4) | 2.78 (±2.5) | 2.72 (±2.4) | 2.75 (±2.5) | 2.63 (±2.4) | 2.67 (±2.3) | 2.72 (±2.3) | 2.64 (±2.3) | 2.70 (±2.5) | 2.71 (±2.6) | 2.68 (±2.5) | 2.72 (±2.5) | 2.72 (±2.6) |
| Inflation-adjusted costs ($), median (IQR) | 11,655 (13,613) | 8,446 (9,227) | 8,291 (9,259) | 9,239 (10,404) | 9,421 (10,592) | 10,057 (10,477) | 10,861 (12,552) | 11,521 (12,722) | 12,095 (13,345) | 12,860 (14,552) | 13,977 (15,565) | 14,689 (16,002) | 15,582 (16,859) | 16,898 (18,241) | 17,756 (19,434) |
| Weekend admission, % | 25.2 | 24.3 | 24.6 | 24.0 | 24.0 | 24.9 | 24.9 | 25.6 | 24.9 | 25.7 | 26.3 | 26.3 | 26.3 | 26.6 | 25.6 |
| Admission quarter, % | | | | | | | | | | | | | | | |
| • Jan—Mar | 27.7 | 28.5 | 28.1 | 27.8 | 27.7 | 29.2 | 25.9 | 28.6 | 29.00 | 27.0 | 28.2 | 24.6 | 27.7 | 27.8 | 28.0 |
| • Apr—Jun | 23.5 | 23.4 | 23.7 | 22.7 | 23.2 | 23.2 | 22.2 | 23.4 | 24.3 | 22.4 | 25.0 | 22.4 | 24.6 | 24.1 | 24.8 |
| • Jul—Sep | 20.5 | 18.9 | 18.7 | 20.3 | 20.4 | 20.4 | 22.8 | 20.2 | 18.5 | 21.5 | 19.6 | 26.1 | 19.5 | 21.2 | 19.9 |
| • Oct—Dec | 28.3 | 29.2 | 29.5 | 29.2 | 28.7 | 27.3 | 29.2 | 27.8 | 28.2 | 29.0 | 27.3 | 26.9 | 28.2 | 26.9 | 27.4 |
| Region of hospital, % | | | | | | | | | | | | | | | |
| • Northeast | 27.7 | 29.4 | 23.9 | 31.2 | 26.9 | 31.5 | 32.1 | 27.9 | 27.6 | 25.5 | 25.8 | 26.0 | 26.2 | 26.8 | 25.5 |
| • Midwest | 21.4 | 22.4 | 23.0 | 22.3 | 25.0 | 22.0 | 19.8 | 22.4 | 24.5 | 19.9 | 19.7 | 21.2 | 19.8 | 18.2 | 18.4 |
| • South | 34.4 | 31.7 | 35.9 | 31.8 | 33.3 | 33.1 | 30.7 | 30.2 | 34.5 | 37.8 | 37.5 | 36.2 | 36.8 | 37.5 | 37.4 |
| • West | 16.5 | 16.5 | 17.1 | 14.7 | 14.8 | 13.4 | 17.5 | 19.5 | 13.3 | 16.8 | 16.9 | 16.7 | 17.3 | 17.5 | 18.7 |

SD: standard deviation; IQR: interquartile range; LOS: length of stay.

Overall, 0.11% of patients admitted with asthma during 2004–2017 died in hospital, with peak of 0.18% of patients in 2017. Annual trends of in-hospital mortality were assessed by age and/or sex, race, asthma severity (S2 Fig). Over time, mean age of death among patients aged <18 ranged between 6–14 years, while it declined in adults from 62 years in 2004 to 53 in 2017. By age and sex, mean age of death in adult patients was 54 in males and 60 in females. By asthma severity, more adults with asthma of extreme severity died than those with less severe disease over (P = 0.000). By sex (across all ages), more females died in hospital then males (P = 0.000), but the proportions of deaths in males tripled from 0.05% to 0.15% during 2004–2017. By race, in-hospital mortality was high among Asian/Pacific Islanders but increased over

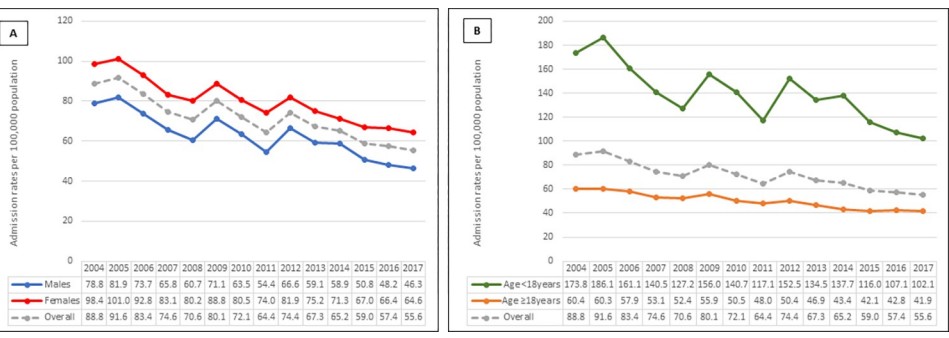

**Fig 1.** Annual rates of asthma admissions by (A) sex and (B) age of patients admitted between 2004 and 2017.

time in White and Black patients. Comparing the observed trends before and after the ICD switch show no apparent impact of the code change on the results as described in S2 and S3 Tables and S3 Fig.

## Annual trends of comorbidity burden

Hypertension, obesity, diabetes, GERD, dyslipidaemia, and CVD were the most prevalent comorbidities during 2004–2017 (S4 Table). Prevalence of depression and psychoses increased by >50% over time. By age, GERD, OSA, anemia, and obesity were prevalent in younger patients (<18years), while diabetes, hypertension, and CVD were prevalent in adults (Fig 2A). By sex, depression (P = 0.000) and hypothyroidism (P = 0.000) were more prevalent in females than males (Fig 2B). By race, hypertension (P = 0.000), diabetes (P = 0.0001) were less common in Hispanic patients than other race groups (Fig 2C).

## Association of comorbidities with outcomes

*Length of stay (LOS).* Asthma admissions between April-December were associated with 1-day shorter hospital spells than January-March admissions (Table 2). Patients with most severe asthma had 1-day longer stays (IRR: 2.48; 2.27–2.72) than patients with minor severity. Age, hypertension, GERD, OSA, chronic sinusitis, and psychoses were predictors of 1-day longer stays.

**Intubation/Ventilator use.** 1-year older patients (OR: 0.99; 95%CI: 0.98–0.99) and females (OR: 0.89; 0.80–1.00) were less likely to receive ventilation over time, whereas, Black (2.01; 1.77–2.28) and Asian/Pacific Islanders (2.35; 1.73–3.20) were more likely to be intubated compared to White (S5 Table). Patients with CVD (0.65; 0.50–0.85) or hypothyroidism (0.60; 0.41–0.89) were at a lower risk. Obese patients admitted with asthma were more likely to receive intubation/ventilation in 2004 (1.27; 1.04–1.56) but less likely in 2017 (0.86; 0.76–0.98) than non-obese patients.

**Total hospital costs.** Costs were significantly higher in females and asthma patients with hypertension (coefficient: 1405.2; 283.1–2527.4) or psychoses (1978.4; 674.9–3282.0) (S6 Table). For concise reporting, we selectively present the associations between comorbidities and outcomes in adult patients only.

## Discussion

### Main findings

We report contemporary national hospitalizations data and systematically examine the burden of comorbidities in asthma patients by patient factors over 14 years. Over time, asthma

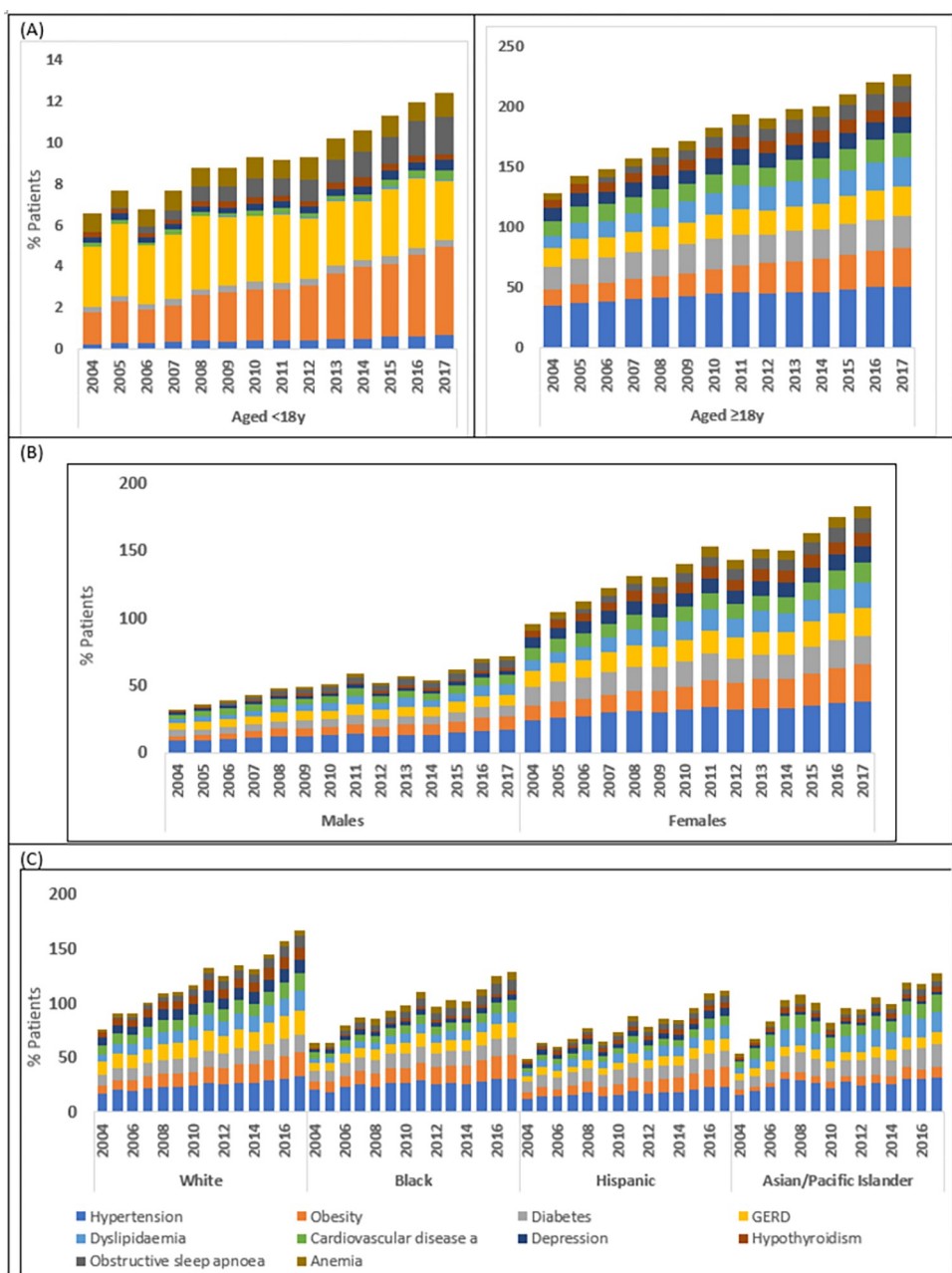

**Fig 2.** Annual weighted trends of comorbidities by (A) age, (B) sex, and (C) race in patients admitted with asthma between 2004 and 2017. GERD: gastroesophageal reflux disease.

admissions started to decline from 2013, whilst ventilator use and in-hospital-deaths increased. Mean age increased over time and most admitted patients were females. Hypertension, obesity, diabetes, GERD, and CVD were the most prevalent comorbidities, while anemia was more prevalent in younger patients. Over time, the prevalence of mental illness increased by >50%. Non-White patients were more likely to receive intubation/ventilation than White. Older patients, with GERD, OSA, or psychoses had longer stays. Despite these changes in demographics towards a more comorbid phenotype, LOS remained unchanged although costs doubled even after inflation-adjustment.

**Table 2. Incidence rate ratios (95% CI) for length of hospital stay (days) in adult patients (18+ years) admitted with asthma in 2004, 2010 and 2017.**

|  | 2004 | 2010 | 2017 |
|---|---|---|---|
| Age | 1.01 (1.01; 1.01) | 1.00 (1.00; 1.05) | 1.00 (1.00; 1.01) |
| Male | Ref | Ref | Ref |
| Female | 1.12 (1.09; 1.14) | 1.10 (1.08; 1.13) | 1.14 (1.10; 1.17) |
| Race |  |  |  |
| White | Ref | Ref | Ref |
| Black | 0.96 (0.94; 0.99) | 0.91 (0.89; 0.93) | 1.02 (0.99; 1.05) |
| Hispanic | 1.00 (0.96; 1.04) | 1.01 (0.97; 1.04) | 1.04 (1.00; 1.09) |
| Asian/Pacific Islander | 0.91 (0.85; 0.96) | 0.93 (0.88; 1.00) | 1.00 (0.94; 1.07) |
| Native American | 0.92 (0.81; 1.05) | 0.99 (0.88; 1.10) | 0.96 (0.85; 1.07) |
| Other | 0.97 (0.88; 1.07) | 1.00 (0.94; 1.06) | 0.99 (0.93; 1.05) |
| Unknown | 0.93 (0.91; 0.95) | 0.91 (0.88; 0.94) | 0.88 (0.81; 0.94) |
| Weekend admission | 0.99 (0.97; 1.01) | 0.98 (0.96; 1.01) | 0.99 (0.96; 1.02) |
| Admission quarter |  |  |  |
| Jan–Mar | Ref | Ref | Ref |
| Apr–Jun | 0.94 (0.91; 0.97) | 0.95 (0.92; 0.97) | 0.97 (0.94; 1.00) |
| Jul–Sep | 0.89 (0.87; 0.92) | 0.93 (0.90; 0.96) | 0.92 (0.89; 0.96) |
| Oct–Dec | 0.95 (0.93; 0.97) | 0.93 (0.91; 0.96) | 0.94 (0.91; 0.97) |
| Elective admission | 1.11 (1.08; 1.14) | 1.04 (1.00; 1.08) | 1.13 (1.07; 1.20) |
| Asthma severity (degree of loss of function) |  |  |  |
| Minor | Ref | Ref | Ref |
| Moderate | 1.24 (1.21; 1.27) | 1.21 (1.18; 1.23) | 1.15 (1.12; 1.19) |
| Major | 1.69 (1.62; 1.76) | 1.57 (1.52; 1.63) | 1.43 (1.39; 1.48) |
| Extreme | 2.63 (2.36; 2.93) | 2.47 (2.27; 2.69) | 2.48 (2.27; 2.72) |
| Comorbidities [a] |  |  |  |
| Diabetes | 0.99 (0.96; 1.01) | 1.00 (0.98; 1.03) | 1.00 (0.97; 1.03) |
| Hypothyroidism | 1.00 (0.97; 1.03) | 1.00 (0.97; 1.03) | 1.05 (1.01; 1.09) |
| Anemias | 1.09 (1.05; 1.13) | 1.12 (1.08; 1.16) | 1.08 (1.02; 1.14) |
| RA/collagen vascular disease | 0.99 (0.92; 1.07) | 1.07 (1.01; 1.14) | 1.08 (1.01; 1.15) |
| Liver disease | 0.99 (0.91; 1.08) | 1.08 (0.97; 1.19) | 1.07 (0.98; 1.16) |
| CKD | 0.98 (0.85; 1.14) | 0.95 (0.90; 1.00) | 1.03 (0.98; 1.09) |
| Psychoses | 1.10 (1.06; 1.16) | 1.10 (1.05; 1.14) | - |
| Depression | 1.07 (1.04; 1.10) | 1.04 (1.02; 1.07) | 1.03 (0.99; 1.06) |
| Weight loss | 1.29 (1.03; 1.60) | 1.26 (1.10; 1.44) | 1.36 (1.15; 1.61) |
| Obesity | 1.03 (1.00; 1.06) | 1.06 (1.04; 1.09) | 1.10 (1.07; 1.13) |
| Chronic sinusitis | 1.19 (1.14; 1.24) | 1.14 (1.09; 1.18) | 1.29 (1.15; 1.45) |
| COPD | 0.98 (0.94; 1.03) | 1.09 (1.03; 1.15) | 1.06 (1.02; 1.10) |
| Obs sleep apnoea | - | 1.07 (1.04; 1.11) | 1.07 (1.04; 1.11) |
| GERD | 1.10 (1.08; 1.13) | 1.12 (1.09; 1.15) | 1.10 (1.07; 1.13) |
| Cancer | 1.09 (0.92; 1.11) | 1.04 (0.96; 1.14) | - |
| Lung cancer | - | - | 0.97 (0.78; 1.20) |
| Dyslipidaemia | 0.96 (0.93; 0.99) | 0.99 (0.96; 1.01) | 0.98 (0.95; 1.01) |
| Coagulopathies | 1.12 (0.98; 1.27) | 1.19 (1.05; 1.36) | 1.21 (1.07; 1.36) |
| Hypertension | 1.04 (1.02; 1.06) | 1.03 (1.01; 1.06) | 1.04 (1.01; 1.07) |
| CVD | 0.98 (0.95; 1.01) | 1.00 (0.97; 1.03) | 1.04 (1.01; 1.08) |

[a] Comorbidities identified as significant predictors for length of hospital stay from univariable regression models per year. RA: rheumatoid arthritis; CKD: chronic kidney disease; COPD: chronic obstructive pulmonary disease; GERD: gastroesophageal reflux disease; CVD: cardiovascular disease.

### Findings in relation to literature

**Admissions trends.**　Our findings show declining rates of US asthma admissions over time, in line with past reports [1, 11]. This reduction is likely related to improvements in asthma management strategies and better asthma care in primary and community health centres; declining US asthma prevalence from 2012 (though followed by a brief increase before decreasing again in 2017) [4, 20]; and the CDC-reported reduction in asthma-related physician office visits from 409.7 per 10,000 population in 2001 to 307.8 per 10,000 population in 2016 [4], which may again be driven by better asthma care, patient education, and self-management interventions. Additionally, we found that the proportion of patients with more severe asthma increased from 0.9% to 2.7% over time while the proportion of admitted patients with least severity level declined from 56% to 42% which possibly indicate that physicians may have developed a higher threshold for admissions where patients with severe asthma exacerbations are prioritised to be admitted resulting in fewer but more severe patients being hospitalised. Reductions in asthma hospitalisations over the last years have also been reported elsewhere including Latin American region, Europe, and the Middle East [11]. Our study includes more years of data providing an update using contemporary data, with the latest previous studies examining US asthma admissions up to 2011 [1, 3]. Kaur et al. reported overall slightly higher US asthma hospitalizations than ours, but both show declining trend in overlapping years (2004–2010) [1]. Our estimated 185,565 admissions (57.4/100,000 population) are, however, very similar to the CDC report of 189,000 (5.9/10,000population) admissions, both in 2016 [4]. The relative proportions of ethnicity are overall close to previous studies, however, our analysis extending to 2017 shows a slightly lower proportion of White (36%) alongside increases in admissions of patients from non-White ethnicities compared to older studies (up to 41%) [1, 10]. We found most admissions were females, as past studies [3, 7, 10, 11]. Previously, females accounted for 75% (2002–2005) [10] and 67% (2001–2010) [1] of US asthma admissions, but we found 57% (2004–2017) indicating a declining gender-gap. As ours, Woods (2010) reported low rates of in-hospital mortality (0.07–0.12%) [10]. We found that most deaths were in Asian/Pacific Islanders. A qualitative study attributed this to dissatisfaction with received care, poor adherence, lack of self-management awareness, and language barrier [21]. Just over 25% of hospitalizations were on weekends, as reported previously [1]. Across study years, asthma admissions were notably higher during fall/winter (October-March), which agrees with reports attributing this to age [22], higher respiratory infections prevalence, and proneness to indoor air pollutants [1, 3, 23].

**Comorbidity burden.**　The identification and management of comorbidities are key recommendations by guidelines and global asthma reports, as they contribute to symptoms burden, and affect asthma control and outcomes [8, 24–27]. GERD, hormonal disturbances, mental illness, COPD, OSA, and obesity are common asthma comorbidities [8, 24]. Obesity is hypothesized to act on asthma through its major role in the development of OSA and GERD and this is in parallel with increasing prevalence of OSA, obesity, and asthma [8]. Our findings are in agreement showing doubling/tripling prevalence of GERD, obesity, and OSA in asthma admissions over time. Additionally, we found high prevalence of comorbidities recently recognized as increasingly associated with asthma, including hypertension, diabetes, CVD [28]. This is important with possible clinical implications, as having >1 comorbidity affect asthma control and recurrent exacerbations [8, 29]. Patients with asthma and mental illness have higher rates of functional impairment and healthcare utilization than patients with either condition alone [28]. Becerra (2016) reported mental illness was associated with increased LOS and costs in US asthma hospitalizations [7]. Similarly, we found depression and psychoses were associated with both outcomes.

**Predictors of in-hospital outcomes.** The need for studies examining the relationship between common asthma comorbidities and asthma outcomes has been acknowledged previously [28]. Non-White ethnicities and having OSA were predictors for in-hospital *intubation/ventilation use*. In agreement with previously-reported predictors of *LOS*, we found age [1] and mental comorbidities [7] were associated with 1-day longer stays; fall-season admissions predictive of slightly shorter stays than winter admissions [1]. Predictors of higher *costs* in our study included age and females [1]; and psychoses [7], as reported previously. Admissions of Hispanic (2001–2010) were reportedly associated with lower costs than Whites [1]. On the contrary, we found that admissions of Hispanic patients (2004–2017) costed more than White patients' admissions.

## Possible clinical and research implications

Asthma and associated hospitalizations are burdensome to healthcare systems [11]. Our findings highlight the changing landscape of sociodemographic and clinical factors in asthma admissions, which may inform redesigning of services to improve pre-hospitalization asthma care. Some symptoms of comorbidities (e.g. breathlessness) may influence the assessment of asthma control leading to unnecessarily-intensified management of asthma, while the symptoms-causing comorbidities need treatment. This highlights the need for national analyses similar to ours to help understand the comorbidity profile in asthma admissions, how it changes over time, and its association with outcomes. Asthma guidelines prioritise identifying comorbidities, particularly those affecting evaluation of disease control and outcomes [8, 24, 26, 27]. Our mapping of comorbidity profile by patient factors and assessing their association with poor in-hospital outcomes can help identify high-risk groups to help inform tailored asthma management, but also highlights the need for future work on identifying comorbidity clusters in people admitted with asthma and examine their prognostic impact on clinical outcomes. Mental illness prevalence almost doubled during our study period and was associated with outcomes. These findings, alongside prior reports on the burden of mental illness in asthma hospitalizations [7], highlight the need for mental health policies and healthcare professionals to promote screening and addressing mental health during routine asthma care. This can help prevent poor outcomes, as psychological factors trigger asthma symptoms and can affect patients' asthma symptom perception and medication adherence [8]. We would expect our findings to be broadly generalizable due to the diversity of the analysed large cohort, in terms of age and ethnicity. With possible hypotheses of how comorbidities affect asthma [8, 28], further research is needed to understand how comorbidities contribute to or interact with asthma.

## Strengths and limitations

Our study has some strengths. First, it is a contemporary analysis of the trends of hospitalizations in patients admitted with asthma up to 2017 using a nationally representative sample of the US population. Second, the study period is longer than previous NIS-based asthma hospitalizations studies [1, 3, 7, 10]. To the best of our knowledge, this is the first study to examine the annual trends of US asthma hospitalizations, patient and hospital factors, comorbidity burden, and in-hospital outcomes categorized by patient factors. While the NIS data presents the largest US inpatient care dataset, some limitations should be considered before interpreting our results. First, inevitably for most administrative databases, there is a possibility of inaccurate or erroneous coding. Second, given the unit of analysis is the admission record, the data may include recurrent hospitalizations of the same patient, which cannot be verified. Third, lack of information on the timing of in-hospital events. Fourth, lack of pharmacological

therapies data, and hence was not considered in the analyses. Fifth, cause of death is not recorded, hence the reported deaths cannot be identified whether caused by asthma and should be interpreted as all-cause mortality. Sixth, the indicator of asthma severity was used as a proxy based on available data in the records and may not be the standard measure for asthma severity driven by several markers such as forced expiratory volume (FEV1), rescue inhaler use, or the frequency of night-time awakenings. Finally, not all possible comorbidities were included, but we limited the analysis to asthma-relevant and verified AHRQ/ELIXHAUSER comorbidities to minimise misclassification.

## Conclusions

Asthma hospitalizations in the US have decreased over time which may reflect improvements in community care. Our temporal findings highlight the changing landscape of asthma admissions, such as increasing comorbidity burden including mental illness, which may inform redesigning asthma services to promote better pre-hospitalization care. Further studies are needed to help understand how comorbidities contribute to or interact with asthma.

## Supporting information

**S1 Table. Annual trends of admissions and baseline characteristics of patients admitted with asthma between 2004 and 2017.**
(PDF)

**S2 Table. Comparison of asthma admissions before and after the ICD code switch by calendar quarters in 2015.**
(PDF)

**S3 Table. Comparison of main patients' characteristics before and after the ICD code switch in 2015.**
(PDF)

**S4 Table. Annual comorbidity profile in patients admitted with asthma between 2004 and 2017.** [a] Cardiovascular disease: any of myocardial infarction, coronary heart disease, heart valve disease, peripheral vascular disease, heart failure, atrial fibrillation/flutter, TIA/stroke. RA: rheumatoid arthritis; COPD: chronic obstructive pulmonary disease; GERD: gastroesophageal reflux disease; TIA: transient ischaemic attack.
(PDF)

**S5 Table. Odds ratio (95% CI) for in-hospital intubation or mechanical ventilation in adult patients (18+ years) admitted with asthma in 2004, 2010 and 2017.** [a] Comorbidities identified as significant predictors for in-hospital intubation/mechanical ventilation from univariable regression models per outcome. RA: rheumatoid arthritis; COPD: chronic obstructive pulmonary disease; CVD: cardiovascular disease.
(PDF)

**S6 Table. Predictors of inflation-adjusted total hospitalization charges for adult patients (18+ years) admitted with asthma in 2004, 2010 and 2017.**
(PDF)

**S1 Fig.** Trends of length of stay (LOS) (panels A, B) and total costs (C, D) by sex and asthma severity of people admitted with asthma between 2004 and 2017.
(PDF)

**S2 Fig. Trends of in-hospital mortality by age of death, gender, race and asthma severity in patients admitted with asthma between 2004 and 2017.**
(PDF)

**S3 Fig. Assessment of asthma admissions before and after the ICD code switch by calendar quarters 2012–2017.**
(PDF)

## Author Contributions

**Conceptualization:** Salwa S. Zghebi, Mamas A. Mamas, Evangelos Kontopantelis.

**Data curation:** Salwa S. Zghebi, Mohamed O. Mohamed.

**Formal analysis:** Salwa S. Zghebi.

**Funding acquisition:** Salwa S. Zghebi.

**Project administration:** Salwa S. Zghebi.

**Supervision:** Mamas A. Mamas, Evangelos Kontopantelis.

**Writing – original draft:** Salwa S. Zghebi.

**Writing – review & editing:** Salwa S. Zghebi, Mohamed O. Mohamed, Mamas A. Mamas, Evangelos Kontopantelis.

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
