## [Decision Letter · Decision Letter 0]

29 Sep 2022

PONE-D-22-22558Temporal trends of hospitalizations, comorbidity burden and in-hospital outcomes in patients admitted with asthma in the United States: population-based studyPLOS ONE

Dear Dr. Zghebi,

Thank you for submitting your manuscript to PLOS ONE. After careful consideration, we feel that it has merit but does not fully meet PLOS ONE’s publication criteria as it currently stands. Therefore, we invite you to submit a revised version of the manuscript that addresses the points raised during the review process.

We look forward to receiving your revised manuscript.

Kind regards,

Sreeram V. Ramagopalan

Academic Editor

PLOS ONE

Journal Requirements:

Reviewers' comments:

Reviewer's Responses to Questions

**Comments to the Author**

1. Is the manuscript technically sound, and do the data support the conclusions?

Reviewer #1: Yes

2. Has the statistical analysis been performed appropriately and rigorously? 

Reviewer #1: Yes

3. Have the authors made all data underlying the findings in their manuscript fully available?

Reviewer #1: Yes

4. Is the manuscript presented in an intelligible fashion and written in standard English?

Reviewer #1: Yes

5. Review Comments to the Author

Reviewer #1: This is a well conducted analysis on a large cohort of patients providing a systematic and comprehensive understanding of changes in the burden of asthma hospitalization and comorbidity in the US over 2004-2017. The research questions are clear and addressed using appropriate methods. The article is well written.

Some minor points:

- The measure of asthma severity is not standard and requires some explanation

- There is some evidence of differences before and after the switch from ICD-9 to ICD-10 in years 2014-2014 (admission rate) and the authors should comment on whether associated differences in coding practices might have meant different types of patients were selected before and after this change and any subsequent biases

- I don't quite understand how fig 2 bars add up to 100% - are patients only allowed one comorbidity?

- The authors suggest that the reduction in hospitalisation rates over time might be due to improved community healthcare - have they considered that physicians have developed a higher threshold for hospital admission - this would result in fewer but more severe patients being admitted - which is consistent with the pattern seen.

6. PLOS authors have the option to publish the peer review history of their article (what does this mean?). If published, this will include your full peer review and any attached files.

Reviewer #1: No

---

## [Author Response · Author response to Decision Letter 0]

11 Oct 2022

Dear Editor,

Re. Submission ref. PONE-D-22-22558

Thank you for the opportunity to submit a revision of our manuscript. Below, we provide a point-by-point response to the received comments. The changes are highlighted on the revised manuscript and numbered by the comment number to facilitate the review process. 

Looking forward to receiving your decision.

Yours sincerely,

S Zghebi, Corresponding author on behalf of the authors.

Journal Requirements

Authors' response

We confirm the manuscript meets the journal's style requirements including file naming.

2) In your Data Availability statement, you have not specified where the minimal data set underlying the results described in your manuscript can be found. PLOS defines a study's minimal data set as the underlying data used to reach the conclusions drawn in the manuscript and any additional data required to replicate the reported study findings in their entirety. All PLOS journals require that the minimal data set be made fully available. For more information about our data policy, please see http://journals.plos.org/plosone/s/data-availability. Upon re-submitting your revised manuscript, please upload your study’s minimal underlying data set as either Supporting Information files or to a stable, public repository and include the relevant URLs, DOIs, or accession numbers within your revised cover letter. For a list of acceptable repositories, please see http://journals.plos.org/plosone/s/data-availability#loc-recommended-repositories. Any potentially identifying patient information must be fully anonymized.

Important: If there are ethical or legal restrictions to sharing your data publicly, please explain these restrictions in detail. Please see our guidelines for more information on what we consider unacceptable restrictions to publicly sharing data: http://journals.plos.org/plosone/s/data-availability#loc-unacceptable-data-access-restrictions. Note that it is not acceptable for the authors to be the sole named individuals responsible for ensuring data access. We will update your Data Availability statement to reflect the information you provide in your cover letter.

Authors' response and change

Access to the data is only granted upon approval by the data owner, the US Healthcare Cost and Utilization Project (HCUP). The Data Availability statement has been updated to: "The data underlying the results presented in the study are available on request from the Healthcare Cost and Utilization Project (HCUP) - National (Nationwide) Inpatient Sample (NIS) (https://www.hcup-us.ahrq.gov/)". This is in line with Data Availability statements in previous PLoS ONE publications based on the same database (DOI: 10.1371/journal.pone.0258345 and DOI: 10.1371/journal.pone.0256757).

Authors' response

We confirm the references are complete and correct and no changes were made to the list. We have not cited papers that have been retracted.

Authors' response

We thank the editor for directing us to this useful digital diagnostic tool. The figures uploaded on this revision have been checked via PACE to ensure they meet PLOS requirements.

Review Comments to the Author

Reviewer #1: This is a well conducted analysis on a large cohort of patients providing a systematic and comprehensive understanding of changes in the burden of asthma hospitalization and comorbidity in the US over 2004-2017. The research questions are clear and addressed using appropriate methods. The article is well written.

Some minor points:

1) The measure of asthma severity is not standard and requires some explanation

Authors' response

Thank you for the positive comments on our study. The NIS dataset provides this variable which describes the 'severity of illness subclass', in this case, asthma. The variable is categorised into the following levels: (0) No class specified, (1) Minor loss of function, (2) Moderate loss of function, (3) Major loss of function, and (4) Extreme loss of function. The admissions included in our analysis were found to be flagged to levels 1-4, none to level '0'. We therefore used this as a surrogate indicator of disease (asthma) severity per admission, although acknowledging it may not be a standard measure for asthma severity.

Changes to the paper

• Methods section - The following text has been edited to further clarify the variable: "… asthma severity (indicated using a categorical variable which describes the 'severity of illness subclass': (0) No class specified, (1) Minor loss of function, (2) Moderate loss of function, (3) Major loss of function, (4) Extreme loss of function),…".

• Discussion section - The following text has also been added to the limitations to highlight that this variable is a proxy based on available information in the database and may not be the standard measure for asthma severity: "Sixth, the indicator of asthma severity was used as a proxy based on available data in the records and may not be the standard measure for asthma severity driven by several markers such as forced expiratory volume (FEV1), rescue inhaler use, or the frequency of night-time awakenings." 

2) There is some evidence of differences before and after the switch from ICD-9 to ICD-10 in years 2014-2014 (admission rate) and the authors should comment on whether associated differences in coding practices might have meant different types of patients were selected before and after this change and any subsequent biases

Authors' response

We have acknowledged the switch from ICD-9 to ICD-10 coding from 1st October 2015 in the 'Study population' section in the paper and we did not observe sharp changes in the annual admissions or prevalence rates. However, the database provider suggests a few approaches to assess the possible effect of the coding change on trends of diagnoses, such as: a) report 4th quarter separately from quarters 1-3 for year 2015; b) report from only one coding scheme. We explore the first approach below, but the second approach is not feasible for our study as we aimed to provide an understanding of the hospitalisation trends over the whole study period 2004-2017. 

Table X1 ICD-9 coding use ICD-10 coding use

 Jan - September 2015

(Calendar quarters 1-3) October 2015

(Calendar quarter 4)

 Q1 Q2 Q3 Full Q 1-3 

Weighted admissions (N) 52,500 46,690 36,995 136,185 53,550

Admissions per 100,000 population 65.3 58.1 46.0 56.5 66.6

Table X1 and Fig X1 show an increase in admissions from 46 to 66.6/100,000 population following the code switch from 4th quarter in 2015. When further explored, we found that the increase in admissions rate between Q3 and Q4 seems consistent in years before and after the code switch i.e. it appears to be unrelated to the code change from Q4 in 2015 (dashed lines in Fig X2). As reported in the paper, asthma admissions were overall higher during fall and winter (October-March), which is in line with previous literature attributing this to higher prevalence of respiratory infections and proneness to indoor air pollutants [1-3].

Regarding the possible effect of the code switch on patient selection, we did not find noticeable changes in the annual trends of patient and hospital characteristics during 2014-2016 (Table 1, Table S1, Table S2 (now S4). However, as justifiably suggested by the reviewer, we assessed if the patients selected before and after the code switch differed, but we found no differences that can be attributed to that switch as shown in Table X2 below.

Table X2 ICD-9 coding use

(Jan 2004 – Sept 2015) ICD-10 coding use

(Oct 2015 – Dec 2017)

Age, mean (±SD) 28.12 (±25) 31.29(±26)

Female, % 56.68 58.62

Ethnicity, %

• White 

35.63 

36.86

• Black * 26.41 32.91

• Hispanic 15.26 18.13

• Asian/Pacific Islander 1.89 2.69

• Native American 0.67 0.73

• Other 3.87 4.51

• Unknown** 16.27 4.16

Asthma severity, %

• Minor 52.92 42.62

• Moderate 37.65 37.11

• Major*** 8.23 17.87

• Extreme 1.19 2.39

*The annual proportion of Black patients was already increasing before the code switch (increased from 23% in 2004 to 32.4% in 2014 i.e. in the ICD-9 phase) (Table 1)

**The annual proportion of patients with unknown ethnicity was already declining before the code switch (declined from 26% in 2004 to 5% in 2014 i.e. in the ICD-9 phase) (Table 1)

***The annual proportion of patients with major severity was already increasing before the code switch (increased from 6.4% in 2004 to 11.1% in 2014 i.e. in the ICD-9 phase) (Table 1)

Changes to the paper

• Methods section – In response to this important comment, we added the following text to further highlight the need to assess possible impact of the code switch in October 2015: "The database provider recommends a few approaches to assess the possible effect of the switch from ICD-9 to ICD-10 coding from 1st October 2015 on trends of diagnoses. We assessed the possible impact of the ICD switch on the trends of asthma admissions by examining whether the patients' characteristics and calculated admission rates differed before and after the code switch by calendar quarters.".

• Results section – we added the new analyses presented above and the following text: "Comparing the observed trends before and after the ICD switch show no apparent impact of the code switch on the results as described in S2 and S3 Tables and S3 Fig.".

3) I don't quite understand how fig 2 bars add up to 100% - are patients only allowed one comorbidity?

Authors' response

Thank you for this comment. Figure 2 represents the weighted percent proportion of each comorbidity per year, displayed as 100% stacked bars. Patients may have more than one comorbidity. We agree with the reviewer that the presented plots may introduce confusion to readers as the 100% stacked bars would display values related to the total per column, hence we have re-plotted Figure 2 as stacked bar chart displaying actual % prevalence of each comorbidity (please see below). Due to the amount of information included in the plots (comparing 10 comorbidities in 2 groups over 14 years), standard bar charts would be illegible, hence the selection of stacked bar charts.

Figure 2 (Panel A) as presented in the previous submission

(A) 

New Figure - Panels A (by age) & B (by sex) 

(A)

(B)

Changes to the paper

Figure 2 has been re-plotted replacing the previous plots as explained above.

4) The authors suggest that the reduction in hospitalisation rates over time might be due to improved community healthcare - have they considered that physicians have developed a higher threshold for hospital admission - this would result in fewer but more severe patients being admitted - which is consistent with the pattern seen.

Authors' response

We thank the reviewer for this valid observation as our findings indeed show that the proportion of patients with more severe asthma increased from 0.9% to 20% over time while the proportion of admitted patients with least severity level declined from 56% to 42% (Table 1). The reduction in US asthma hospitalisation rates have been reported previously [1, 4, 5] and we believe this is possibly attributed to several causes including improved asthma care, declining asthma prevalence from 2012 onwards [5, 6]. The US CDC also report that the rate of asthma-related physician office visits declined from 409.7 per 10,000 population in 2001, to 307.8 per 10,000 population in 2016 [5], which again may be driven by better asthma care, patient education, and self-management interventions. In addition, as the reviewer pointed out, development of higher threshold for admissions where patients with severe asthma exacerbations are likely prioritised to be admitted, can ultimately contribute to the observed reduction in hospitalisation rates.

Changes to the paper

Discussion section - The text has been modified in response to this comment and now reads as follows: "Our findings show declining rates of US asthma admissions over time, in line with past reports [1, 4]. This reduction is likely related to improvements in asthma management strategies and better asthma care in primary and community health centres; declining US asthma prevalence from 2012 (though followed by a brief increase before decreasing again in 2017) [5, 6]; and the CDC-reported reduction in asthma-related physician office visits from 409.7 per 10,000 population in 2001 to 307.8 per 10,000 population in 2016 [5], which may again be driven by better asthma care, patient education, and self-management interventions. Additionally, we found that the proportion of patients with more severe asthma increased from 0.9% to 20% over time while the proportion of admitted patients with least severity level declined from 56% to 42% which possibly indicate that physicians may have developed a higher threshold for admissions where patients with severe asthma exacerbations are prioritised to be admitted resulting in fewer but more severe patients being hospitalised. Reductions in asthma hospitalisations over the last years have also been reported elsewhere including Latin American region, Europe, and the Middle East [4].".

1. Kaur BP, Lahewala S, Arora S, Agnihotri K, Panaich SS, Secord E, et al. Asthma: Hospitalization Trends and Predictors of In-Hospital Mortality and Hospitalization Costs in the USA (2001–2010). Int Arch Allergy Immunol. 2015;168:71–8. doi: 10.1159/000441687.

2. Mehal JM, Holman RC, Steiner CA, Bartholomew ML, Singleton RJ. Epidemiology of Asthma Hospitalizations Among American Indian and Alaska Native People and the General United States Population. CHEST. 2014;146(3):624 - 32. doi: 10.1378/chest.14-0183.

3. McCoy L, Redelings M, Sorvillo F, Simon P. A Multiple Cause-of-Death Analysis of Asthma Mortality in the United States, 1990–2001. Journal of Asthma. 2005;42(9):757-63. doi: 10.1080/02770900500308189.

4. Cabrera A, Rodriguez A, Romero-Sandoval N, Barba S, Cooper PJ. Trends in hospital admissions and mortality rates for asthma in Ecuador: a joinpoint regression analysis of data from 2000 to 2018. BMJ Open Respiratory Research. 2021;8(1):e000773. doi: 10.1136/bmjresp-2020-000773.

5. The Centers for Disease Control and Prevention (CDC). Asthma Prevalence and Health Care Resource Utilization Estimates, United States, 2001-2017. 2017.

6. Lucas JA, Marino M, Fankhauser K, Bazemore A, Giebultowicz S, Cowburn S, et al. Role of social deprivation on asthma care quality among a cohort of children in US community health centres. BMJ Open. 2021;11(6):e045131. doi: 10.1136/bmjopen-2020-045131.

---

## [Editor Report · Decision Letter 1]

13 Oct 2022

Temporal trends of hospitalizations, comorbidity burden and in-hospital outcomes in patients admitted with asthma in the United States: population-based study

PONE-D-22-22558R1

Dear Dr. Zghebi,

We’re pleased to inform you that your manuscript has been judged scientifically suitable for publication and will be formally accepted for publication once it meets all outstanding technical requirements.

Kind regards,

Sreeram V. Ramagopalan

Academic Editor

PLOS ONE
---

## [Editor Report · Acceptance letter]

17 Nov 2022

PONE-D-22-22558R1 

Temporal trends of hospitalizations, comorbidity burden and in-hospital outcomes in patients admitted with asthma in the United States: population-based study 

Dear Dr. Zghebi:

I'm pleased to inform you that your manuscript has been deemed suitable for publication in PLOS ONE. Congratulations! Your manuscript is now with our production department. 

Kind regards, 

on behalf of

Dr. Sreeram V. Ramagopalan 

Academic Editor

PLOS ONE